# Position: Stop Blaming Aggregation in Deep GNNs Based on Incomplete Evidence

## Abstract

This position paper argues that aggregation mechanisms in Graph Neural Networks (GNNs) have been unfairly blamed for limiting network depth based on incomplete evidence. The prevailing narrative, shaped by early observations of best performance at shallow depths and theoretical concepts such as oversmoothing and oversquashing, has led researchers to view aggregation as fundamentally problematic for deep GNNs. However, through reexamining the experimental evidence and theoretical explanations by proper ablations, we show that aggregation's negative effects have been exaggerated while its benefits have been overlooked. Based on our findings, we advocate two actions: First, practitioners should actively deploy deep GNNs on tasks that require large capacity or long-range dependencies, whose potential has already been demonstrated yet remains overlooked. Second, researchers should rigorously investigate the true effects of aggregation on deep GNNs rather than accepting conventional wisdom.

## 1. Introduction

Graph Neural Networks (GNNs) have become a central modeling paradigm for structured data, achieving strong empirical performance across a wide range of domains, including social networks (Wang et al., 2019a; Li & Goldwasser, 2019; Huang et al., 2019; Fan et al., 2019), molecules (Gilmer et al., 2017; Zitnik & Leskovec, 2017; Dai et al., 2019), knowledge graphs (Wang et al., 2019b;c; Park et al., 2019), and recommendation systems (Monti et al., 2017; Ying et al., 2018; Jiang et al., 2022; Wu et al., 2022b). A key source of this success lies in their message passing mechanism, which incorporates neighboring information from the graph structure. The message passing mechanism is primarily re-

alized through the aggregation operation, which is one of the three fundamental components of GNN architectures: aggregation, transformation, and activation. Since the transformation and activation operations are structurally identical to those used in multilayer perceptrons (MLPs), aggregation is the defining component that makes GNNs a powerful extension of standard neural networks.

A large body of research has focused on the role of aggregation and its effects on representation learning in GNNs. Early work (Li et al., 2018) has shown that common aggregation schemes can be interpreted as a special form of Laplacian smoothing, encouraging neighboring nodes to learn similar representations, which can be beneficial when neighboring nodes share the same label. They also show that, however, the infinitely repeated application of such smoothing results in oversmoothing, a regime in which node representations become indistinguishable. Related phenomena, such as oversquashing (Topping et al., 2022; Deac et al., 2022), have also been introduced to describe limitations arising from information compression during message passing. Together, these studies have provided valuable insights into the mathematical and empirical properties of aggregation in GNNs.

These aggregation-centered explanations, combined with the observation reported in the celebrated GCN paper (Kipf & Welling, 2017) that the best results are obtained with a 2- or 3-layer model, fostered the view that GNNs inherently prefer shallow depth, and performance degrades as depth increases. The subsequent introduction of demonstration of oversmoothing's exponential occurrence (Oono & Suzuki, 2020; Cai & Wang, 2020; Wu et al., 2023) even reinforces the view by explaining how oversmoothing can affect performance in relatively shallow depth. As a result, the perspective has become widespread across the field, leading researchers to primarily consider shallow architectures when designing GNNs. Indeed, our examination of publicly available GNN models reveals that hyperparameter search spaces for network depth are predominantly restricted to fewer than ten layers (Bo et al., 2021; He et al., 2021; Luo et al., 2024; Pei et al., 2020; Platonov et al., 2023).

It is important to note, however, that several works have demonstrated the successful construction of deep GNNs. Li

[1]Anonymous Institution, Anonymous City, Anonymous Region, Anonymous Country. Correspondence to: Anonymous Author <anon.email@domain.com>.

Preliminary work. Under review by the International Conference on Machine Learning (ICML). Do not distribute.

et al. (2019) achieved state-of-the-art performance on point cloud semantic segmentation (Armeni et al., 2017) using a 56-layer GNN. Li et al. (2021) trained a 448-layer GNN that achieved the best performance on the large-scale ogbn-proteins dataset (Hu et al., 2020). These results clearly indicate that depth itself is not an insurmountable obstacle for GNNs when properly addressed. Notably, neither work focused primarily on mitigating aggregation-induced problems. Their solutions targeted classical deep learning challenges: vanishing gradients and memory efficiency, respectively.

Despite these advances, the narrative that building deep GNNs is fundamentally difficult has persisted. Based on the aggregation-as-obstacle belief, many studies aimed at enabling deeper architectures focus specifically on mitigating oversmoothing (Zhao & Akoglu, 2020; Rong et al., 2020; Chen et al., 2020; Zhou et al., 2021b; Eliasof et al., 2021; Chamberlain et al., 2021; Thorpe et al., 2022; Rusch et al., 2022; Fang et al., 2023; Song et al., 2023; Rusch et al., 2023b; Park et al., 2024a; Su et al., 2024; Pei et al., 2024; Lu et al., 2024). The persistence of this narrative, even in the face of successful deep GNN constructions, reflects the compelling nature of the foundational evidence: empirical observations of shallow-depth optimality coupled with theoretically rigorous aggregation-centered explanations.

However, we argue that a fundamental oversight exists in this evidence: GNNs are, at their core, structured around MLP components. Overfitting in low-capacity regimes and gradient instability, patterns identical to those observed in deep GNNs, are well documented in the broader deep learning literature as inherent to deep MLPs. Since GNNs share the same transformation and activation layers as MLPs, they likely inherit these identical optimization hurdles. Nevertheless, prior work has implicitly blamed aggregation for these challenges without conducting the necessary ablations to isolate message passing from standard MLP dynamics.

Motivated by this observation, we revisit the empirical foundations of deep GNN optimization by disentangling the individual contributions of transformation and activation steps. Through a systematic re-examination of experimental evidence and theoretical frameworks, we demonstrate that the purported negative effects of aggregation have been substantially overstated. Our findings reveal that GNNs actually exhibit greater robustness to increased depth than their MLP counterparts, directly contradicting the prevailing view that aggregation is the primary driver of performance degradation. Theoretically, we show that existing oversmoothing analyses entail a critical yet overlooked implication: learned transformations can actively counteract convergence. Furthermore, we provide evidence that observed exponential oversmoothing is primarily driven by transformation and activation steps, with aggregation's contribution being relatively marginal.

Based on these findings, **we argue that aggregation has been unfairly maligned for deep GNN challenges based on incomplete evidence**. We advocate two actions. First, practitioners should actively deploy deep GNNs for tasks that require high capacity or long-range dependencies. Second, researchers should rigorously investigate the actual effects of aggregation on optimization and generalization. We present our evidence in Section 3, outline these directions in Section 4, and address credible counterarguments in Section 5.

## 2. Backgrounds

The dominant narrative that aggregation in GNNs inherently limits network depth has been constructed through a series of influential yet incomplete observations. This narrative began with early empirical findings from two highly influential works: Graph Convolutional Networks (GCN) (Kipf & Welling, 2017) and Graph Attention Networks (GAT) (Veličković et al., 2018), both of which reported best performance at shallow depths of 2-3 layers on the Cora, CiteSeer, and PubMed datasets (Sen et al., 2008).

Both architectures share a common structure. At each layer $\ell$, node features are updated as:

$$\mathbf{X}^{(\ell+1)} = \sigma(\hat{\mathbf{A}}\mathbf{X}^{(\ell)}\mathbf{W}^{(\ell)}), \tag{1}$$

where $\mathbf{X}^{(0)}$ are input features, $\sigma(\cdot)$ is typically ReLU, $\mathbf{W}^{(\ell)}$ are learnable weights, and $\hat{\mathbf{A}}$ is the aggregation operator (a degree-normalized adjacency matrix in GCN; a learnable right-stochastic attention matrix in GAT). The single layer comprises three sequential steps: *aggregation* ($\tilde{\mathbf{X}}^{(\ell+1)} = \hat{\mathbf{A}}\mathbf{X}^{(\ell)}$), *transformation* ($\tilde{\mathbf{X}}^{(\ell+1)} = \hat{\mathbf{X}}^{(\ell+1)}\mathbf{W}^{(\ell)}$), and *activation* ($\mathbf{X}^{(\ell+1)} = \sigma(\tilde{\mathbf{X}}^{(\ell+1)})$). We note that removing aggregation yields a standard multilayer perceptron (MLP) ($\mathbf{X}^{(\ell+1)} = \sigma(\mathbf{X}^{(\ell)}\mathbf{W}^{(\ell)})$), while Simple Graph Convolution (SGC) (Wu et al., 2019) removes activation, applying multiple aggregations followed by a single transformation: $\mathbf{X}^{(L)} = \hat{\mathbf{A}}^L\mathbf{X}^{(0)}\mathbf{W}$.

Among these three steps, Li et al. (2018) identified aggregation as the source of depth limitations. They introduced the concept of *oversmoothing*, showing that GCN's aggregation is a special form of Laplacian smoothing. When repeated infinitely, aggregation causes node features resulting in the degree-scaled embedding convergence. Several studies reinforce the claim by proving an asymptotic oversmoothing regime in various forms of aggregation from different perspectives. Keriven (2022) demonstrate that the repeated aggregation process with the right stochastic matrix results in the uniform embedding convergence, using the ergodic theorem. Chamberlain et al. (2021) interpret GATs as a discretization of the heat diffusion equation, implying that

the embedding reaches equilibrium. Based on this intuition, Thorpe et al. (2022) provide a theoretical analysis showing that the outputs of GATs asymptotically converge to a fixed value independent of the initial inputs. However, such asymptotic analyses do not explain why performance degradation occurs in GNNs within the non-asymptotic regime, that is, when GNNs have only a small number of layers.

Subsequent work (Oono & Suzuki, 2020; Cai & Wang, 2020; Wu et al., 2023) seem to answer the question, providing a theoretical analysis that the oversmoothing occurs at an exponential rate. To quantify the speed of oversmoothing, these studies first define a node similarity measure $\mu : \mathbb{R}^{N \times d} \to \mathbb{R}_{\geq 0}$ that approaches zero when node embedding converges. With the proposed measure, the exponential oversmoothing can be shown as an exponential decay of the measure, such that

$$\mu(\mathbf{X}^{(\ell)}) \leq C q^{\ell} \text{ for } \ell = 1, \cdots, L - 1, \qquad (2)$$

for some constant $C > 0$, and $0 < q < 1$. In Appendix B, we summarize four representative node similarity measures, $\mu_{\text{oono}}$, $\mu_{\text{cai}}$, $\mu_{\text{wu}}$, and $\mu_{\text{rusch}}$ from Oono & Suzuki (2020); Cai & Wang (2020); Wu et al. (2023); Rusch et al. (2023a), respectively. By addressing a question left by previous asymptotic analyses, this line of research has further strengthened the theoretical foundation for the belief that aggregation introduces inherent limitations in deep GNNs.

These early observations and persuasive explanations led to the widespread belief that GNNs inherently suffer significant performance degradation as depth increases (Oono & Suzuki, 2020; Zhao & Akoglu, 2020; Zhou et al., 2021b;a; Zhang et al., 2022; Thorpe et al., 2022; Song et al., 2023). This belief was further solidified by the identification of the over-squashing problem, which suggests that the aggregation structure inherently limits the model's ability to learn long-range dependencies. Since then, many studies have either examined the intrinsic problems of aggregation in deep GNN construction more rigorously or attempted to construct deep GNNs and improve their performance by addressing these issues. We summarize these lines of work in Appendix A.

However, some studies provide findings that challenge such established views. For example, Li et al. (2019) successfully trained a 56-layer GCN by adapting skip connections. Similarly, Li et al. (2021) achieve the best performance using a 448-layer GNN, primarily by addressing memory consumption issues. Zhou et al. (2021a) and Zhang et al. (2022) show no significant performance degradation when repeatedly applying the aggregation process without the transformation step. Some studies (Yang et al., 2020; Cong et al., 2021) show that oversmoothing does not occur after training. Moreover, Rusch et al. (2023a) and Heo et al.

(2025) show that node similarity measures show empirically weak correlation with performance.

## 3. Revisiting the Evidence

The narrative that aggregation limits depth rests mainly on two sources: depth-performance experiments (Kipf & Welling, 2017; Veličković et al., 2018) showing optimal results at shallow layers, and theoretical analysis explaining performance degradation via oversmoothing (Li et al., 2018; Oono & Suzuki, 2020; Chamberlain et al., 2021; Thorpe et al., 2022; Wu et al., 2023). In this section, we revisit these evidence through systematic ablations and careful review of theoretical claims.

### 3.1. Depth-Performance Patterns

We evaluate the training and test accuracy of vanilla GCN, GAT, and MLP while varying the depth from 1 to 64 on Cora, CiteSeer, and PubMed datasets used in Kipf & Welling (2017) and Veličković et al. (2018). Detailed experimental setups are provided in Appendix F. The results are illustrated in Figure 1. Dashed and solid lines represent training and test accuracy, respectively.

The experimental results show that GNNs, specifically GCN and GAT, exhibit better robustness to increased network depth than MLPs. MLPs suffer sharp declines in both training and test accuracy at depths of 8 or 16, likely due to vanishing gradients. In contrast, GCN and GAT maintain relatively stable performance up to depth 32, with comparable drastic degradation occurring only at greater depths. Notably, GNNs also show narrower gaps between training and test accuracy as depth increases, suggesting aggregation may improve generalization. These results provide a counterexample to the conventional wisdom that aggregation-induced issues, such as oversmoothing or oversquashing, cause significant performance degradation as depth increases. Rather, this ablation study with MLPs reveals a finding that has received little attention: *aggregation may contribute to robustness of performance as depth increases.*

Despite this robustness advantage, both GCNs and GATs achieve their best test performance at shallow depths of two or four layers on these datasets, consistent with prior observations. However, we observe that MLPs also achieve their best performance at two or four layers. Moreover, across all three datasets, all three models generally maintain high training accuracy close to 100% before vanishing gradient.

This suggests that the tasks do not require the additional capacity that depth provides. In such settings, increased depth primarily introduces overfitting risk rather than performance gains. Therefore, to examine whether GNNs inherently prefer shallow depths, evaluation on tasks that demand greater capacity would be necessary. Indeed, GNNs have

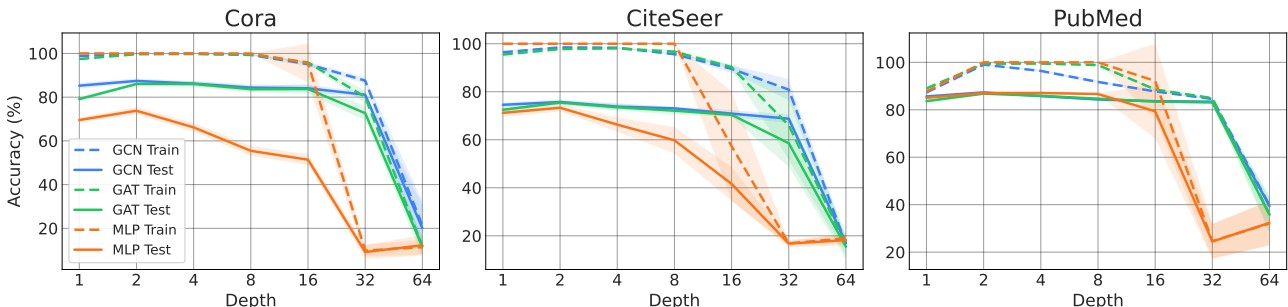

*Figure 1.* Training and test accuracy of GCN, GAT, and MLP across varying depths (1-64 layers) on three citation network datasets: Cora, CiteSeer, and PubMed. Dashed lines represent average training accuracy and solid lines represent average test accuracy over 5 repeated runs. Shaded regions indicate the standard deviation around the average. All three models achieve best test performance at shallow depths (2-4 layers) and maintain high training accuracy, suggesting these tasks do not require the capacity that depth provides. Notably, GNNs maintain stable performance to depth 32 while MLPs suffer drastic degradation beyond depth 16, indicating aggregation provides robustness even when depth does not improve peak performance on these specific tasks.

been shown to benefit from increased depth on large-scale datasets (Li et al., 2021) and tasks that require capturing long-range dependencies (Park et al., 2024a;b), which we discuss further in Section 4.

### 3.2. Theoretical Analysis

Oversmoothing has been widely recognized as a fundamental obstacle to deep GNN construction. This belief has been solidified by rigorous theoretical analyses and their empirical validations demonstrating oversmoothing's occurrence in GNNs. However, considering the transformation process in GNNs raises an overlooked question: *Can learned transformations prevent oversmoothing?* Intuitively, the answer should be yes. Unless aggregation makes representations perfectly identical, transformations can amplify differences and offset smoothing, even through simple scalar multiplication. Aligned with this intuition, some studies (Yang et al., 2020; Cong et al., 2021) observe that oversmoothing does not occur after training, suggesting weight parameters learn to offset smoothing effects. This observation motivates closer examination of what conditions the theory actually requires for oversmoothing to occur.

We found that this phenomenon aligns with the theoretical analysis provided by Oono & Suzuki (2020) and Cai & Wang (2020). For example, in GCN, Oono & Suzuki (2020) demonstrate

$$\mu_{\text{oono}}(\mathbf{X}^{(\ell)}) \leq (s\lambda)^{\ell} \mu_{\text{oono}}(\mathbf{X}^{(0)}) , \tag{3}$$

where $s$ is the largest operator norm of $\mathbf{W}^{(\ell)}, \forall \ell$, $\lambda < 1$ is the second largest eigenvalue of $\hat{\mathbf{A}}_{\text{GCN}}$. Based on Equation (3), the exponential oversmoothing occurs only when the combined effects of aggregation and transformation drive the node similarity measure toward zero, i.e., $s\lambda < 1$. The theory does not imply that oversmoothing is inevitable; rather, occurrence depends on how weight parameters

evolve during training. Since cross-entropy loss encourages separating representations of different classes, we can expect transformations to learn in directions that counteract aggregation's smoothing effect.

Similarly, Cai & Wang (2020) demonstrate $\mu_{\text{cai}}(\mathbf{X}^{(\ell)}) \leq (s(1 - \bar{\lambda})^2)^{\ell} \mu_{\text{cai}}(\mathbf{X}^{(0)})$ where $\bar{\lambda}$ indicates smallest non-zero eigenvalue of augmented normalized Laplacian $\mathbf{I}_N - \hat{\mathbf{A}}_{\text{GCN}}$. Despite using a different node similarity measure and a slightly different exponential base, the presence of the weight-dependent term $s$ leads to the same conclusion: *oversmoothing is conditional on learned parameters, not inevitable*.

Not all theoretical work has carefully accounted for weight parameters. Some analyses (Li et al., 2018; Chamberlain et al., 2021; Keriven, 2022; Thorpe et al., 2022) focus on oversmoothing in the asymptotic regime of SGC structures, where aggregation is repeated infinitely without intermediate transformations or activations ($\lim_{L \to \infty} \hat{\mathbf{A}}^L \mathbf{XW}$), thereby sidestepping the question of how learned weights interact with aggregation across layers. Wu et al. (2023) demonstrates exponential oversmoothing in GATs, explicitly considering the transformation step in their analysis. However, in deriving upper bounds on node similarity measures, they treat the weight-dependent term as a constant and factor it out of the exponential. This results in bounds that appear independent of weight parameters, overlooking how transformation influences the exponential rate of oversmoothing.

### 3.3. Empirical Observation on Exponential Oversmoothing

While the theory's conditional nature suggests oversmoothing is not inevitable with proper optimization, a critical concern remains: *what if oversmoothing occurs so rapidly in initialized networks that optimization cannot even begin?*

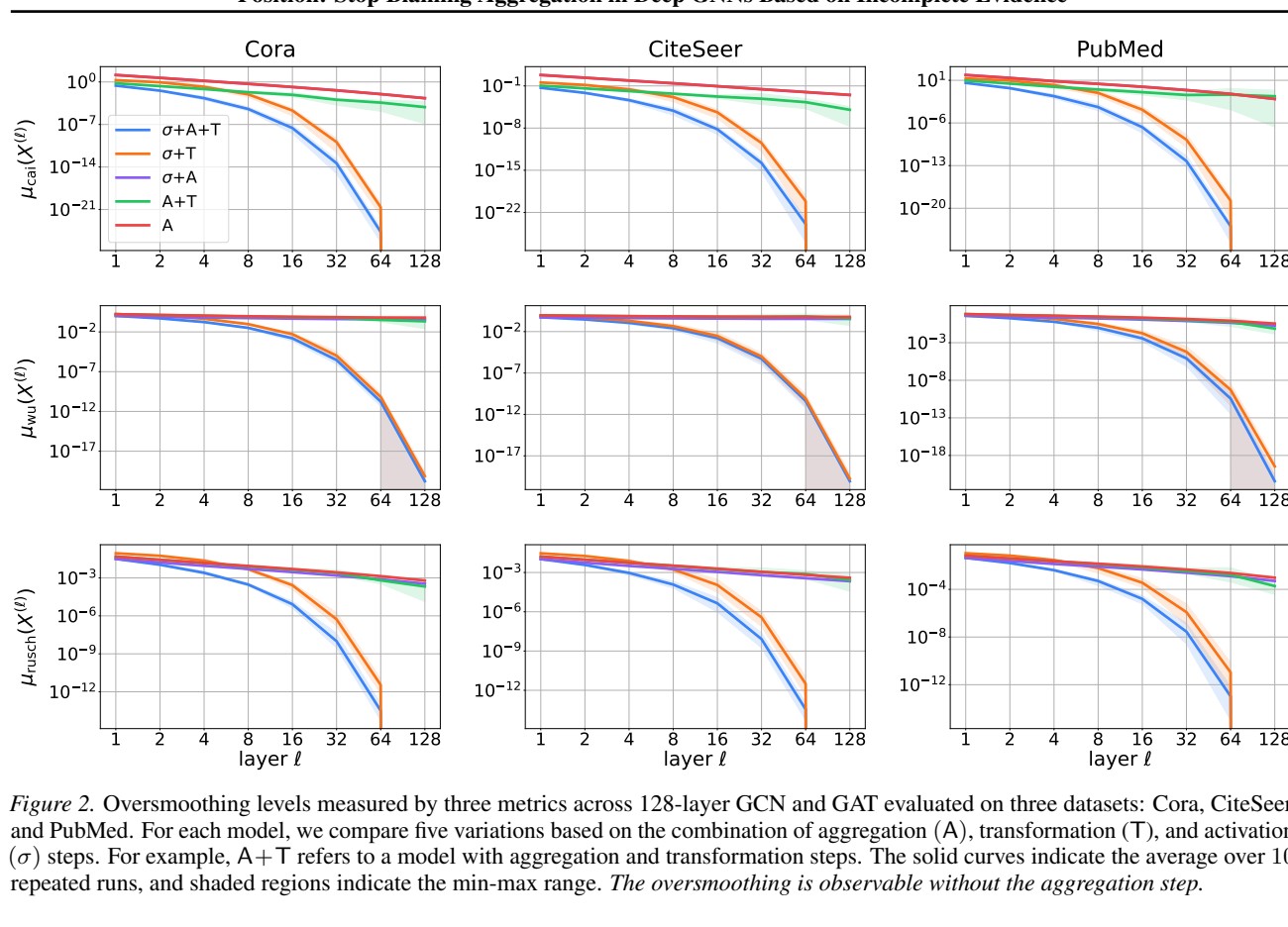

*Figure 2.* Oversmoothing levels measured by three metrics across 128-layer GCN and GAT evaluated on three datasets: Cora, CiteSeer, and PubMed. For each model, we compare five variations based on the combination of aggregation (A), transformation (T), and activation ($\sigma$) steps. For example, A+T refers to a model with aggregation and transformation steps. The solid curves indicate the average over 10 repeated runs, and shaded regions indicate the min-max range. *The oversmoothing is observable without the aggregation step.*

If node representations converge to identical values before any training takes place, gradients cannot flow properly and there is no signal to guide weight updates. This would render the theoretical escape route via learned transformations practically inaccessible.

Experimental validations of exponential oversmoothing theory appear to confirm this concern. Multiple studies (Oono & Suzuki, 2020; Wu et al., 2023; Rusch et al., 2023a) measure oversmoothing and observe exponential decay approaching zero even at moderate depths around 64 layers. These results have been interpreted as strong evidence that aggregation severely limits practical depth.

However, none of the previous studies report the influence of individual steps on the oversmoothing empirically. Our systematic ablation studies show that the rapid decay cannot be mainly attributed to aggregation. Instead, it is dominated by the combination of transformation and activation, not aggregation.

To quantify the independent influence of individual steps, we measure and compare the changes in node similarity using four variants of the GNNs: the original model ($\sigma$+A+T), the model with activation and transformation ($\sigma$+T), the model with activation and aggregation ($\sigma$+A), the model

with aggregation and transformation (A+T), the model with aggregation-only (A). For example, the model with activation and transformation ($\sigma$+T) effectively becomes an MLP by excluding the aggregation step. The model with aggregation (A) only can be seen as the SGC. We compute node similarity measures from 128-layer GCN and GAT models at depths $\ell = 1, 2, 4, \cdots, 128$ on Cora, CiteSeer, and PubMed datasets: $\mu_{\text{cai}}(\mathbf{X}^{(\ell)})$ for GCN, and $\mu_{\text{wu}}(\mathbf{X}^{(\ell)})$ and $\mu_{\text{rusch}}(\mathbf{X}^{(\ell)})$ for GAT.

Figure 2 shows the changes in the similarity measure over the layers. Regardless of the dataset and the node similarity measure, rapid exponential decay reaching near-zero values around depth $64 \sim 128$ is observed only when both transformation and activation are present (i.e., $\sigma$+A+T, $\sigma$+T). All other configurations, models excluding activation (i.e., A+T), models excluding transformation (i.e., $\sigma$+A), and aggregation-only models (i.e., A), exhibit significantly lower decay rate. Although a mild level of oversmoothing is observed with the aggregation-only model (i.e., A), the decay rate is much lower than that of models including activation.

From these results, we conclude that *the influence of the aggregation step is marginal* and the oversmoothing is primarily caused by the combination of transformation and activation steps. The slow decay rate induced by aggre-

gation can be attributed to graph sparsity: sparser graphs exhibit lower decay rates (Rong et al., 2020), and real-world graphs are typically highly sparse (e.g., Cora has sparsity of $10^{-3}$). These results do not contradict the theoretical analyses of Cai & Wang (2020) and Oono & Suzuki (2020), which attribute exponential oversmoothing to the combination of aggregation and transformation, nor Wu et al. (2023), which attributes it to aggregation and activation. Indeed, exponential decay does occur in these combinations, albeit at much slower rates than in models including both transformation and activation. However, both the theoretical analyses and empirical validations in these studies overlooked the dominant role of transformation and activation in driving rapid exponential decay.

Then why do transformation and activation steps cause oversmoothing? Note that $\sigma + \mathsf{T}$ is equal to the MLP architecture, and we already have a well-known case where the oversmoothing occurs with MLPs: the *zero-collapsing*, whereby node embeddings converge toward the zero vector (Glorot & Bengio, 2010).[1] This occurs most prominently when Glorot (Xavier) initialization (Glorot & Bengio, 2010), originally designed for linear or tanh activations, is paired with ReLU. In the deep learning community, zero-collapsing has been primarily understood as a source of vanishing gradients rather than as oversmoothing, and researchers have developed effective solutions, most notably He initialization (He et al., 2015), which scales weights appropriately for ReLU.

Note that we initialize the weight parameters using Glorot (Xavier) initialization (Glorot & Bengio, 2010) to reproduce experimental setup of previous studies (Oono & Suzuki, 2020; Wu et al., 2023; Rusch et al., 2023a).[2] To show how the results change with different initialization choices, we provide the results using He initialization in Appendix D. The results show that the decay rates of both the original model ($\sigma + \mathsf{A} + \mathsf{T}$) and the model with activation and transformation ($\sigma + \mathsf{T}$) significantly decrease and node similarity measure remain far from zero even at depth 128.

In summary, the rapid exponential oversmoothing previously attributed to aggregation is, in fact, zero-collapsing, a phenomenon that occurs even in architectures without ag-

---

[1]While neuron saturation describes similar behaviors in sigmoid and hyperbolic tangent activations, we employ the term 'zero-collapsing' to denote the specific scenario where activations collapse toward zero, a phenomenon commonly observed with ReLU-like functions.

[2]One puzzling aspect we notice is that official implementations of GNNs provided by several libraries, such as Pytorch Geometric (Fey & Lenssen, 2019) and Chainer (Akiba et al., 2017; Tokui et al., 2015; 2019), adopt Glorot initialization by default, despite ReLU being the most popular activation function in GNNs. We find that numerous studies (Chen et al., 2020; Oono & Suzuki, 2020; Rong et al., 2020; Zhao & Akoglu, 2020; Zhou et al., 2021b; Rusch et al., 2022; 2023a;b; Park et al., 2024a) also commonly adopt Glorot initialization in their official implementations.

gregation, such as MLP. Returning to our earlier question, "*what if oversmoothing occurs so rapidly in initialized networks that optimization cannot even begin?*", we find that while such rapid convergence does occur, it is not caused by aggregation. Consequently, the issue can be resolved through classical solutions such as proper weight initialization (e.g., He initialization), without requiring GNN-specific architectural innovations. If aggregation-induced oversmoothing were rapid enough to cause problematic convergence, such standard solutions would be insufficient. However, our observations show that aggregation induces oversmoothing at a substantially slower rate, slow enough that it poses no concern at the depths typically considered in practice.

**Remark.** We found that the visualization from Li et al. (2018), showing node embeddings becoming indistinguishable as depth increases on Zachary's Karate Club network (Zachary, 1977), also conflates zero-collapsing with aggregation-induced smoothing. We reproduced this experiment with SGC and MLP baselines. The results reveal that MLPs exhibit more severe smoothing than SGC, demonstrating that the observed convergence in GCN was primarily driven by zero-collapsing rather than aggregation. Details are provided in Appendix E.

## 4. Call to Action: What We Recommend Going Forward

Our analysis reveals that the field has been constrained by an incomplete understanding of aggregation's role in deep GNNs. Moving forward requires two complementary approaches: immediate practical deployment of deep GNNs where they are needed, and rigorous theoretical investigation to understand when and why they work. We address each audience in turn.

### 4.1. For Practitioners: Deploy Deep GNNs

**Deep GNNs already work.** Despite the prevailing narrative that aggregation fundamentally limits depth, a parallel line of research has demonstrated successful deep GNN construction. ResGCNs (Li et al., 2019) achieve state-of-the-art performance on point cloud semantic segmentation with 56 layers by employing skip connections to address optimization challenges. RevGNN (Li et al., 2021) solve memory complexity to train 448-layer GNNs, achieving top performance on the large-scale ogbn-proteins dataset. Park et al. (2024a) introduce a reverse GNN structure that makes representations distinguishable, reaching peak performance at 256 layers on several heterophilic datasets. Park et al. (2024b) address the redundant message flow, also known as the backtracking problem, reporting best performance at depth 20 on the Peptides-struct dataset from the long-range

graph benchmark (LRGB) (Dwivedi et al., 2022). We note that 20 was the maximum depth explored in their hyperparameter search.

**Task-dependent depth requirements.**   These cases reveal an important pattern: deep GNNs succeed on specific types of tasks. Large-scale datasets like ogbn-proteins benefit from the increased model expressivity that depth provides. Heterophilic datasets and long-range graph benchmarks require the extended receptive fields that only deep architectures can provide (Rusch et al., 2023a). However, this does not mean depth is universally beneficial. Small-scale datasets like Cora and CiteSeer, may continue to perform best with shallow models.

**Practical recommendations.**   For practitioners working with large-scale datasets, long-range dependency tasks, or heterophilic graphs, we recommend actively exploring deep architectures. Adopting techniques from previous success cases, such as skip connections and memory-efficient training methods, can facilitate deep GNN construction. Depth should be treated as an explorable hyperparameter, not an inherent limitation.

### 4.2. For Researchers: Advance Deep GNNs

**Understanding the real challenges.**   While deep GNNs can be deployed successfully, our understanding of how aggregation actually affects deep architectures remains incomplete. Accurately characterizing aggregation's properties is essential for advancing deep GNNs.

Our suggestion for the first step is identifying *when smoothing becomes problematic*. Current notions of oversmoothing and node similarity measures (Oono & Suzuki, 2020; Cai & Wang, 2020; Wu et al., 2023; Rusch et al., 2023a) focus on convergence of representations. While they capture complete convergence, they provide no guidance on when representations become problematically similar. Yet even modest smoothing, far from full convergence, can be harmful if it mixes representations of different classes.

For identification, we need explicit connections between smoothing and its consequences for learning, such as optimization and generalization performance. Intuitively, smoothing affects optimization directly: it eases gradient descent when representations of the same class become similar, but impedes it when representations of different classes get mixed. Smoothing also impacts generalization, as recent work reveals that clustered intra-class representations and separable inter-class representations are critical for generalization (Chuang et al., 2021). Rigorously capturing when smoothing helps versus harms optimization and generalization would provide inspiration for deep GNN's development.

**Building effective solutions.**   With accurate understanding of aggregation's properties, the field can develop more powerful deep GNN architectures that practitioners can deploy with confidence. We identify two promising directions for developing such architectures.

First, adapt proven classical techniques to work optimally with GNN structures. While skip connections, batch normalization, and He-initialization enable deep GNN training, these techniques were designed without consideration of GNN architecture. Several studies have adapted such classical deep learning techniques to the graph domain, including skip connections (Zhang et al., 2022; Lu et al., 2024), normalization layers (Cai et al., 2021; Zhou et al., 2021a), and initialization (Li et al., 2023), yielding improvements. Further developing these classical techniques based on an accurate understanding of aggregation's properties, and systematically exploring how to optimally integrate the solutions, would significantly advance practical deep GNN design.

Second, develop solutions that address GNN-specific challenges in layer stacking. Methods proposed to mitigate oversmoothing (Zhao & Akoglu, 2020; Rong et al., 2020; Chen et al., 2020; Zhou et al., 2021b; Eliasof et al., 2021; Chamberlain et al., 2021; Thorpe et al., 2022; Rusch et al., 2022; Fang et al., 2023; Song et al., 2023; Rusch et al., 2023b; Park et al., 2024a; Su et al., 2024; Pei et al., 2024; Lu et al., 2024) are valuable when harmful smoothing dominates and needs to be slowed down. This is particularly relevant for heterophilic graphs, which require long-range propagation but where aggregation is likely to mix representations of different classes. Similarly, addressing phenomena like backtracking, which shows stronger performance correlation (Park et al., 2024b), can yield improvements. The key is building frameworks that guide which solutions to apply for specific graph characteristics and task requirements, enabling practitioners to construct deep GNNs effectively without defaulting to shallow architectures.

## 5. Alternative Views

While our position challenges prevailing beliefs about aggregation in deep GNNs, we acknowledge credible alternative perspectives. We address three main counterarguments.

### 5.1. "These Issues Are Already Being Addressed"

**The counterargument.** One might argue that our position is redundant, that the field has already moved beyond viewing aggregation as fundamentally limiting. After all, numerous recent works have successfully constructed deep GNNs (Li et al., 2019; 2021; Park et al., 2024a;b), and several studies have questioned the severity of oversmoothing (Roth & Liebig, 2024; Yang et al., 2020; Cong et al., 2021). If researchers are already building deep GNNs and questioning

aggregation-centric explanations, what novel contribution does our position provide?

**Our response.** While we acknowledge these important developments, they represent fragmented insights rather than a systematic reframing of the field's understanding. The mainstream discourse continues to treat deep GNN construction as fundamentally difficult due to aggregation-induced issues. Major surveys (Rusch et al., 2023a; Ju et al., 2024; Jin & Zhu, 2025), papers (Li et al., 2018; Alon & Yahav, 2021; Wenkel et al., 2022; Giraldo et al., 2023; Qureshi et al., 2023; Wu et al., 2023; Li et al., 2024; Pei et al., 2024; Errica et al., 2025; Sun et al., 2025), and textbooks (Ma & Tang, 2021; Wu et al., 2022a) still prominently feature oversmoothing and oversquashing as inherent limitations of GNN depth, not as problems that have been adequately addressed or questioned. This narrative persists in shaping research priorities and practitioner beliefs.

We argue that these insights failed to become mainstream precisely because they did not directly address the evidentiary foundations, the experimental results and theoretical analyses, that established the aggregation-as-obstacle narrative in the first place. Our contribution is to provide this systematic reexamination of the evidence, unifying fragmented insights into a coherent narrative that can reshape the field's mainstream understanding. We demonstrate through controlled ablation studies that aggregation provides robustness as depth increases, contrary to the widespread belief (Section 3.1), and that exponential oversmoothing dominates by zero-collapsing rather than aggregation (Section 3.3). We also clarify that the theoretical analysis does not imply that oversmoothing is inevitable, but escape with proper optimization (Section 3.2). By directly addressing the evidentiary gaps that previous work left intact, we aim to shift the field from fragmented corrections to a coherent understanding: aggregation's effects are complex and context-dependent, not uniformly limiting, and research should focus on understanding *when* and *why* depth helps rather than assuming it inherently hurts.

### 5.2. "Aggregation Is Indeed Problematic"

**The counterargument.** The success of deep GNNs seems to validate aggregation-centric concerns, since their method addresses aggregation-related issues. Park et al. (2024a) achieves 256-layer depth on heterophilic graphs by modifying aggregation's smoothing property. Non-backtracking GNNs (Park et al., 2024b) improve performance by eliminating message revisitation through the same edges. If fixing aggregation-related issues yields improvements, doesn't this prove aggregation is indeed problematic?

**Our response.** We do not claim that aggregation presents no challenges. Our position is that aggregation's actual effects must be accurately understood. In heterophilic graphs,

smoothing is indeed problematic, yet the oversmoothing regime fails to adequately characterize this, since it focuses on complete convergence without considering class information. This is why we argue in Section 4.2 that research should shift focus toward capturing when smoothing becomes problematic.

Backtracking has also received comparatively little attention within the current oversmoothing- and oversquashing-centered discourse. If future investigations reveal that backtracking impacts deep GNN construction more strongly than oversmoothing or oversquashing, as initial evidence suggests (Park et al., 2024b), it should be prioritized accordingly. In sum, we advocate redirecting research effort toward aggregation properties that demonstrably affect performance. This enables developing targeted solutions for real obstacles rather than overstated limitations.

### 5.3. "Oversmoothing Is Indeed Problematic"

**The counterargument.** Theoretical work (Scholkemper et al., 2024) shows that skip connections and normalization mitigate oversmoothing, suggesting Li et al. (2019)'s success in constructing deep GNNs precisely by solving oversmoothing problems. If fixing oversmoothing issues yields improvements, doesn't this prove oversmoothing is indeed problematic?

**Our response.** We argue that the success of these techniques actually reinforces our position. Skip connections and normalization are foundational deep learning practices, not GNN-specific innovations developed to address a unique "aggregation crisis." If standard methods readily enable deep GNN construction, it suggests that depth limitations were never as fundamental or aggregation-specific as previously claimed. Consequently, the most productive research direction is not to continue obsessing over the problems these classic solutions already solve, but to understand the new architectural landscape they allow us to explore.

## 6. Conclusion

This position paper challenges the narrative that aggregation fundamentally limits depth in GNNs. Through careful re-examination of experimental evidences and theoretical explanation, we demonstrate that: (1) GNNs are more robust to depth than MLPs, (2) theoretical analyses show oversmoothing is not inevitable, and (3) exponential oversmoothing is primarily driven by zero-collapsing from transformation and activation, not aggregation. These findings directly call for two actions. Practitioners should explore deep architectures for high-capacity and long-range tasks. Researchers should accurately characterize aggregation's effects. By moving from incomplete narratives to an accurate understanding, we can unlock the full potential of deep GNNs.

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

# A. Additional Related work

Many studies have attempted to enable deeper stacking of layers without performance degradation by mitigating over-smoothing. PairNorm (Zhao & Akoglu, 2020) maintains total pairwise embedding distances constant through normalization. DropEdge (Rong et al., 2020) randomly removes edges from the graph, thus delaying oversmoothing. Additionally, DropMessage (Fang et al., 2023) unifies and improves random dropping in GNNs by directly masking elements in the message matrix, achieving better generalization, stability, and over-smoothing mitigation. Energetic Graph Neural Networks (Zhou et al., 2021b) incorporate Dirichlet energy into the loss function as a regularizer. Ordered GNN (Song et al., 2023) separately preserves aggregated embeddings within specific hops. Gradient Gating (Rusch et al., 2023b) introduces a mechanism to stop learning individually for each node before local oversmoothing occurs. Park et al. (2024a) propose the framework that can reverse the aggregation process. GCNII (Chen et al., 2020) introduces initial residual connections and identity mapping to solve oversmoothing. LSGAT (Su et al., 2024) adaptively adjusts attention coefficients across layers to avoid oversmoothing. MTGCN (Pei et al., 2024) identifies the root cause of oversmoothing as information loss due to heterophily mixing in aggregation and mitigates it by separating message passing into multiple semantic tracks. SkipNode (Lu et al., 2024) alleviates over-smoothing and gradient vanishing in deep GCNs by selectively skipping message passing for certain nodes. Some studies propose to change the dynamics of GNNs, given that classical GNNs resemble diffusion-like dynamics (Chamberlain et al., 2021). GRAND++ (Thorpe et al., 2022) uses a source term to prevent the convergence of the embedding. PDE-GCN (Eliasof et al., 2021) and Graph-coupled oscillator networks (Rusch et al., 2022) are designed based on the wave function and nonlinear oscillator structures, respectively. However, only a few methods have succeeded in achieving performance improvements with increasing depth.

# B. Node similarity measures

In this section, we provide four node similarity measures used in previous works for completeness and how the exponential oversmoothing is characterized via the similarity measures.

In the literature, we identify two distinct definitions of oversmoothing, each specific to GCNs and GATs, respectively. The following two propositions clarify the distinction between the oversmoothing in GCNs and GATs.

**Proposition B.1** (degree-scaled embedding convergence (Cai & Wang, 2020; Li et al., 2018; Oono & Suzuki, 2020)). *In GCNs, there exist $\mathbf{c} \in \mathbb{R}^d$ such that $\lim_{\ell \to \infty} \mathbf{x}_i^{(\ell)} = \sqrt{d_i + 1}\mathbf{c}$ for all nodes $i \in \mathcal{V}$*

**Proposition B.2** (uniform embedding convergence (Keriven, 2022; Thorpe et al., 2022; Wu et al., 2023)). *In GATs, there exist $\mathbf{c} \in \mathbb{R}^d$ such that $\lim_{\ell \to \infty} \mathbf{x}_i^{(\ell)} = \mathbf{c}$ for all nodes $i \in \mathcal{V}$*

These propositions imply that unless all node degrees are equal, i.e., $d_i = d$ for all $i$, or embeddings converge to the zero vector, i.e., $\lim_{\ell \to \infty} \mathbf{x}_i^{(\ell)} = [0, 0, \cdots, 0]^\top$, GCNs and GATs exhibit distinct behaviors in the asymptotic regime.

Oono & Suzuki (2020) uses distance between embedding $\mathbf{X}$ and a space $\mathcal{M} = \{\tilde{\mathbf{D}}^{\frac{1}{2}}\mathbf{1}_N \otimes \mathbf{w} | \mathbf{w} \in \mathbb{R}^d\} \in \mathbb{R}^{N \times d}$ as a node similarity measure, i.e.:

$$\mu_{\text{oono}}(\mathbf{X}) := \inf\{\|\mathbf{X} - \mathbf{Y}\|_F \,|\, \mathbf{Y} \in \mathcal{M}\}, \tag{4}$$

where $\mathbf{1}_N = [1, \cdots, 1]^\top \in \mathbb{N}^N$, $\otimes$ denotes Kronecker product, and $\|\cdot\|_F$ indicates Frobenius norm. Embedding $\mathbf{X}$ spans on $\mathcal{M}$, i.e., $\mu_{\text{oono}}(\mathbf{X}) = 0$, if and only if $\mathbf{X}$ satisfies Proposition B.1.

For demonstration of the exponential occurrence of oversmoothing in GCN, they show that

$$\mu_{\text{oono}}(\mathbf{X}^{(\ell)}) \leq (s\lambda)^\ell \mu_{\text{oono}}(\mathbf{X}^{(0)}), \tag{5}$$

where $s$ is the largest operator norm of $\mathbf{W}^{(\ell)}, \forall\ell$, $\lambda < 1$ is the second largest eigenvalue of $\hat{\mathbf{A}}_{\text{GCN}}$. In the process of demonstration, they show how each aggregation, transformation, and activation step influences the oversmoothing as follows:

$$\mu_{\text{oono}}(\hat{\mathbf{A}}_{\text{GCN}}\mathbf{X}) \leq \lambda\mu_{\text{oono}}(\mathbf{X}), \ \mu_{\text{oono}}(\mathbf{X}\mathbf{W}) \leq \|\mathbf{W}\|_2 \mu_{\text{oono}}(\mathbf{X}), \ \mu_{\text{oono}}(\sigma(\mathbf{X})) \leq \mu_{\text{oono}}(\mathbf{X}), \tag{6}$$

where $\|\cdot\|_2$ denotes operator norm. Equation (6) suggests that exponential oversmoothing occurs due to the aggregation and transformation step.

Empirical results show that $\mu_{\text{oono}}(\mathbf{X}^{(\ell)})$ exponentially decays in practice, satisfying the inequality provided by the theory. However, an adequate explanation of why the distance continues to decay even in cases where the upper bound explodes due to the condition $s\lambda > 1$ was not provided.

Cai & Wang (2020) simplify the proof of Oono & Suzuki (2020) by using Dirichlet energy as a measure of oversmoothing, i.e.,

$$\mu_{\mathtt{cai}}(\mathbf{X}) := \frac{1}{2} \sum_{i \in \mathcal{V}} \sum_{j \in \mathcal{N}_i} \left\| \frac{\mathbf{x}_i}{\sqrt{1 + d_i}} - \frac{\mathbf{x}_j}{\sqrt{1 + d_j}} \right\|_F^2 . \tag{7}$$

$\mu_{\mathtt{cai}}(\cdot) = 0$ also holds if and only if Proposition B.1. They prove that $\mu_{\mathtt{cai}}(\mathbf{X}^{(\ell)}) \leq (s(1 - \bar{\lambda})^2)^\ell \mu_{\mathtt{cai}}(\mathbf{X}^{(0)})$ by showing:

$$\mu_{\mathtt{cai}}(\hat{\mathbf{A}}_{\mathrm{GCN}}\mathbf{X}) \leq (1 - \bar{\lambda})^2 \mu_{\mathtt{cai}}(\mathbf{X}), \ \mu_{\mathtt{cai}}(\mathbf{XW}) \leq \|\mathbf{W}\|_2 \, \mu_{\mathtt{cai}}(\mathbf{X}), \ \mu_{\mathtt{cai}}(\sigma(\mathbf{X})) \leq \mu_{\mathtt{cai}}(\mathbf{X}), \tag{8}$$

where $\bar{\lambda}$ indicates smallest non-zero eigenvalue of augmented normalized Laplacian $\mathbf{I}_N - \hat{\mathbf{A}}_{\mathrm{GCN}}$. The demonstration process also implies that the aggregation and transformation step causes the exponential oversmoothing.

Wu et al. (2023) extend the previous theoretical analyses to demonstrate that GATs also lose expressive power exponentially. They define a node similarity measure:

$$\mu_{\mathtt{wu}}(\mathbf{X}) := \left\| \mathbf{X} - \frac{1}{N} \mathbf{1}_N \mathbf{1}_N^\top \mathbf{X} \right\|_F , \tag{9}$$

where $\mu_{\mathtt{wu}}(\mathbf{X}) = 0$ if and only if Proposition B.2 holds, instead of Proposition B.1. They show that $\mu_{\mathtt{wu}}(\mathbf{X}^{(\ell)})$ approaches zero at an exponential rate as passing the GAT layers, explaining that the exponential oversmoothing occurred due to the joint spectral radius of a $\hat{\mathbf{A}}_{\mathrm{GAT}}$ being less than one. They validate their theory by numerical experiments, showing that $\mu_{\mathtt{wu}}(\mathbf{X}^{(\ell)})$ exponentially converges to zero in practice.

Rusch et al. (2023a) introduce one more node similarity measure:

$$\mu_{\mathtt{rusch}}(\mathbf{X}) := \sqrt{\frac{1}{N} \sum_{i \in \mathcal{V}} \sum_{j \in \mathcal{N}_i} \|\mathbf{x}_i - \mathbf{x}_j\|_F^2} , \tag{10}$$

which is also called Dirichlet energy but has a different form with $\mu_{\mathtt{cai}}(\mathbf{X})$. $\mu_{\mathtt{cai}}(\cdot)$ satisfies $\mu_{\mathtt{wu}}(\mathbf{X}) = 0$ if and only if Proposition B.2 holds. Although the measure was not utilized in theoretical analysis, Rusch et al. (2023a) provide the experimental results that $\mu_{\mathtt{rusch}}(\mathbf{X}^{(\ell)})$ converges exponentially to zero when $\ell$ increases in GCN, GAT, and GraphSAGE (Hamilton et al., 2017).

## C. Dataset statistics

We provide detailed statistics and explanations about the dataset used for the experiments in Table 1 and the paragraphs below.

*Table 1.* Statistics of the datasets utilized in the experiments.

| Dataset | # nodes | # edges | # features | # classes |
| --- | --- | --- | --- | --- |
| Cora | 2,708 | 5,278 | 1,433 | 7 |
| CiteSeer | 3,327 | 4,552 | 3,703 | 6 |
| PubMed | 19,717 | 44,324 | 500 | 3 |
| Karate Club | 34 | 156 | 34 | 4 |

**Cora, CiteSeer, and PubMed (Sen et al., 2008).** Each node represents a paper, and an edge indicates a reference relationship between two papers. The task is to predict the research subjects of the papers.

**Karate Club (Zachary, 1977).** Each node represents a member of a karate club and an edge indicates a social interaction between two members. The task is to predict the final faction each member belongs to after the club divides into groups.

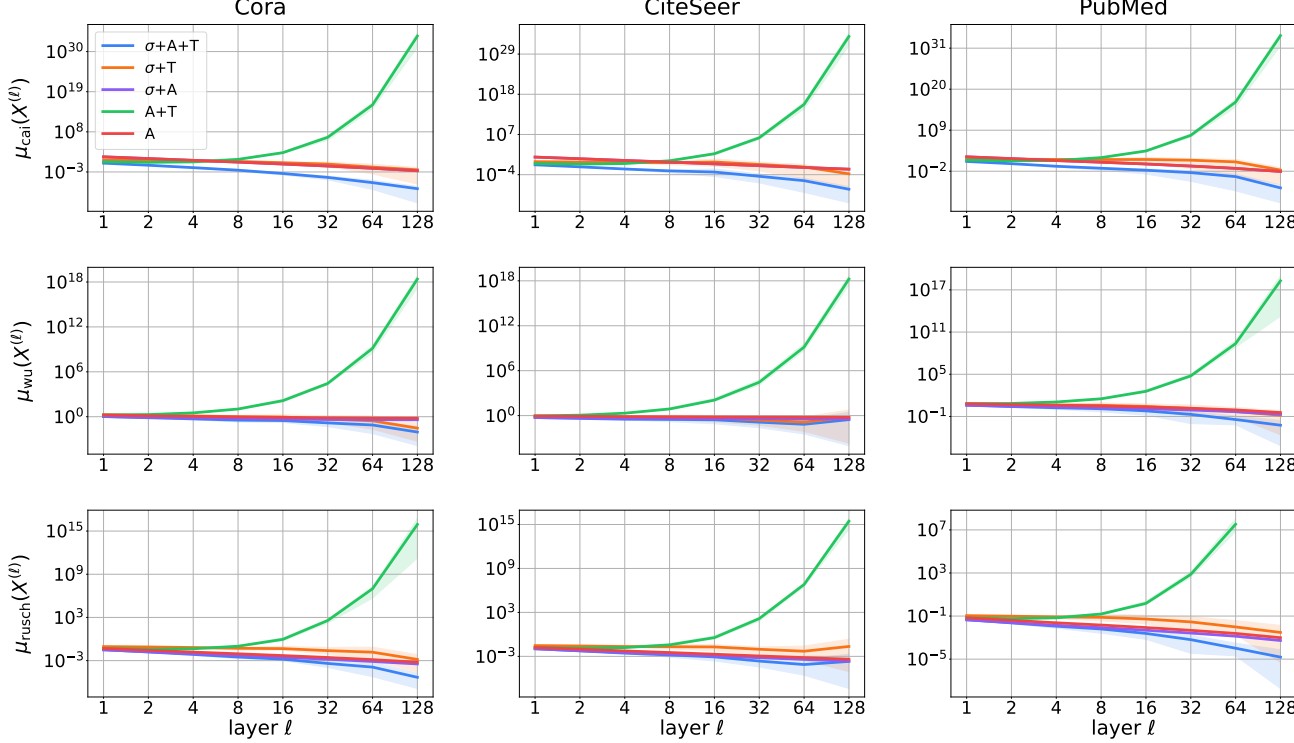

*Figure 3.* Oversmoothing levels, measured by three metrics across 128-layer GNN model with He initialization, evaluated on three datasets: Cora, CiteSeer, and PubMed. For each model, we compare five variations based on the combination of aggregation (A), transformation (T), and activation ($\sigma$) steps. For example, A+T refers to a model with only aggregation and transformation steps. The solid curves indicate the average over 5 repeated runs, and shaded regions indicate the min-max range.

## D. Supplementary experimental results with He initialization

We demonstrate that He initialization can mitigate exponential oversmoothing in GNNs. We compute node similarity measures from 128-layer GCN and GAT models at depths $\ell = 1, 2, 4, \cdots, 128$ on Cora, CiteSeer, and PubMed datasets: $\mu_{\text{cai}}(\mathbf{X}^{(\ell)})$ for GCN, and $\mu_{\text{wu}}(\mathbf{X}^{(\ell)})$ and $\mu_{\text{rusch}}(\mathbf{X}^{(\ell)})$ for GAT. Figure 3 shows the changes of the node similarity measure across layers in GCN and GAT, respectively, with He initialization adopted. We observe that the decay rate significantly decreases compared to results with Glorot initialization shown in Figure 2.

## E. Supplementary visualization of Oversmoothing

Li et al. (2018) provided a visualization showing node embeddings becoming indistinguishable as depth increases on Zachary's Karate Club network using randomly initialized GCNs. We reproduced this experiment with SGC (aggregation only) and MLP (no aggregation) baselines. Rather than forcing two-dimensional outputs as in the original study, we used t-SNE to visualize the high-dimensional embeddings.

Figure 4 reveals that MLPs smooth embeddings more severely than SGC. As discussed in Section 3.3, the smoothing induced by MLPs is due to zero-collapsing. The original study, lacking this baseline comparison, inadvertently attributed zero-collapsing effects to aggregation. This visual ablation reinforces our quantitative findings: aggregation-induced mixing occurs at a much slower rate, remaining visually distinguishable even at depths where MLPs have collapsed.

## F. Experimental Details

In this section, we describe the details of our training setup for experiments in Section 3.1 and Section 3.3. Our experiments were conducted on an AMD EPYC 7513 32-core processor and a single NVIDIA RTX A5000 or RTX 3090 GPU with 24GB of memory.

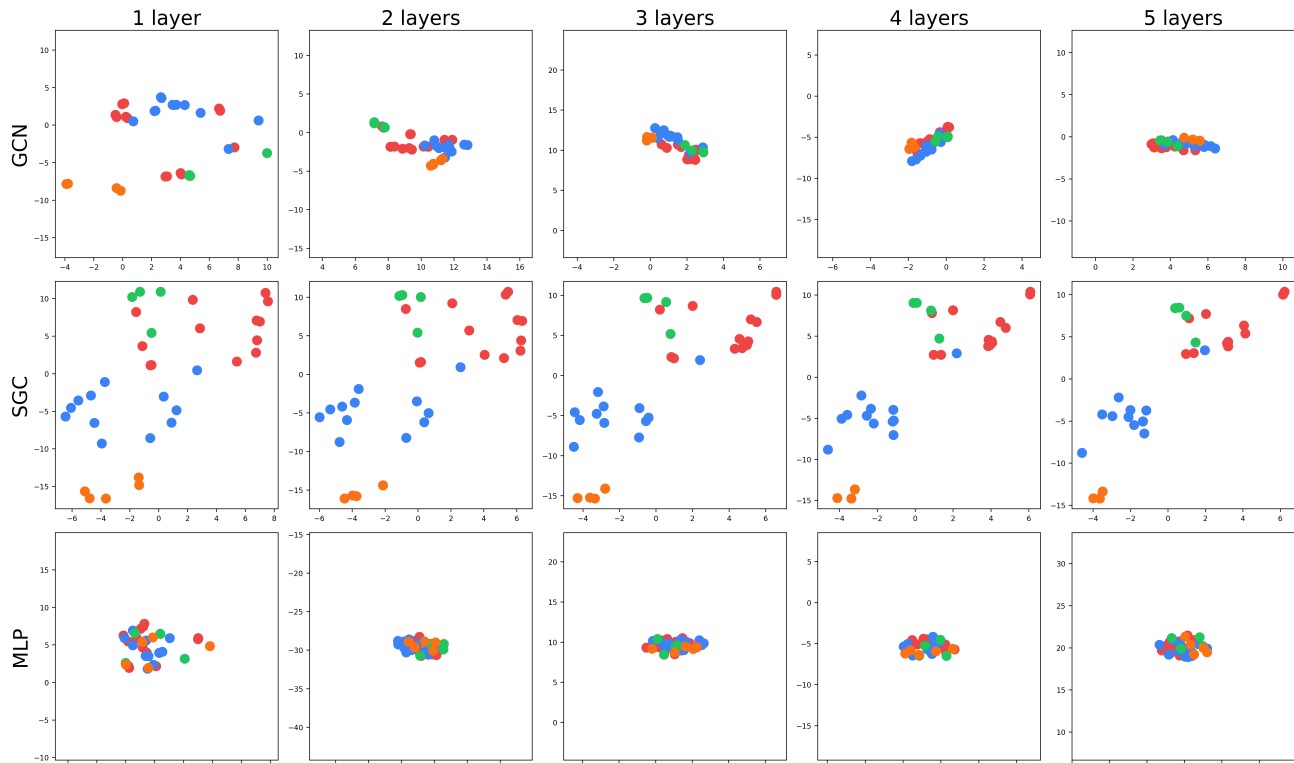

*Figure 4.* t-SNE visualization of node embeddings on Zachary's Karate Club network across varying depths (1-5 layers) for GCN, SGC (aggregation only), and MLP (no aggregation). Each point represents a node, colored by its ground-truth class (4 classes). Contrary to expectations, MLPs exhibit more severe feature mixing than GCNs or SGC, with embeddings collapsing toward a central region due to dying ReLU effects. This demonstrates that the visual mixing phenomenon cannot be attributed solely to aggregation.

To obtain results shown in Figure 1 in Section 3.1, we use the 5 existing standard train/validation/test splits for all datasets. We configure GAT with a single attention head. We train for a maximum of $2,000$ epochs using early stopping on the validation set with a patience of $1,000$ epochs. For each model, we perform a learning rate search within $\{0.0005, 0.001, 0.005, 0.01\}$ and depth search within $\{1, 2, 4, \cdots, 64\}$. We fix the hidden dimension to $64$.

To obtain the results presented in Figure 2 in Section 3.3, we measure node similarity without training, since a 128-layer GNN cannot be trained due to the vanishing gradient issue described in the main paper. Following Wu et al. (2023), we set the hidden dimension to 32 and we use a single-head configuration for GAT.

