# OpenReview forum: "Position: Stop Blaming Aggregation in Deep GNNs Based on Incomplete Evidence"
_ICML.cc/2026/Position_Paper_Track — Submitted to ICML 2026 Position Paper Track_

### Official Review · Reviewer_miFn · 2026-03-10

**Significance:** 2
**Argument Clarity:** 3
**Rating:** 2
**Confidence:** 4

**Questions:**

* How is this position different from the existing papers I mentioned?

* You observe that hyperparameter searches in prior work are "predominantly restricted to fewer than ten layers" and treat this as evidence of a harmful bias. Do you have any evidence that deeper architectures do work better for these datasets?

**Alternative Views Section:**

Yes

**Compliance With Llm Reviewing Policy A Conservative:**

Affirmed.

**Discussion Potential:**

2

**Final Justification:**

After going through the last reply by the authors I admit that [0] does attribute gradient vanishing to the normalized adjacency and that this paper's claim about aggregation not being the issue is a different point. I retract that criticism and agree with the authors here (my last comment about this was a bit of a stretch and I'm grateful for the authors pointing this out). However, this does not invalidate the other points that are still open. The novelty is still very incremental (again, reviewer eAFA raises a similar point with more references). The recommendations the authors make regarding the deployment of deep GNNs are still very simplistic and/or do not need the observations made in this paper (like using "skip-connections and memory-efficient training"). The fact that the paper mainly tests untrained networks (for the main results in Figure 2) further raises the question whether aggregation might still be problematic during training (in the last reply the authors claim that aggregation is exonerated). The paper itself shows that what they observe is mainly an issue of initialization with Glorot instead of He. How this behaves on datasets that require deeper networks is also still not clear (no such results have been provided). The last reply has mainly focused on how this position is different from [0], which I agree with, but not approached the other remaining problems. The position itself may be valid, but the paper only provides thin support and overstretches its claims too much. Finally, I don't think that the paper in its current state will spark discussions in the field and needs to be improved/extended for that.

**Paper Summary:**

The paper argues that oversmoothing in deep GNNs should not be attributed to the aggregation mechanism but that GNNs suffer from the same problems as other deep networks. The authors demonstrate this by using deep MLPs which exhibit a similar behavior and showing that it occurs for GNNs even without aggregation. The paper claims to “systematically re-examine” the evidence which provides novel insights.

**Position:**

Yes

**Position In Title:**

Yes

**Related Work:**

1

**Strengths And Weaknesses:**

Overall, the argument that "aggregation has been unfairly maligned for deep GNN challenges based on incomplete evidence" is valid, but the paper completely misses more recent works that have already shown this and provided evidence (both experimental and theoretical) that GNNs suffer from similar issues as other deep or recurrent architectures. Most notably, [0] does a similar but much more in-depth analysis. The paper is not even cited, just like [1] which makes similar observations. Without even setting the paper into context to these previous works which already provide a similar position, it is hard to see what kind of new perspective this paper provides.

Some other points:

* The paper argues that shallow networks have been shown to work better than deeper ones and vaguely attributes this to the fact that "oversmoothing" was problematic (independent of what causes that). I would argue that the authors somewhat overlook that most GNN datasets do not require deeper GNNs as they are trying to solve local problems (even if the graphs are "large-scale"). This is definitely the case for the datasets used in the experiments (Cora, CiteSeer and PubMed) and the main reason why deeper GNNs are not often used and only for specific tasks/datasets (where they already are used as the authors note). The point that the paper tries to make of deploying deep GNNs everywhere because they “already work” should does be reflected a bit more deeply.

* Purple lines seem to be missing in the first row of Figure 2.

* The paper implicitly argues for the position that research still "prominently feature[s] oversmoothing and oversquashing as inherent limitations of GNN depth", which needs more support, as many papers have shown that deep architectures can work. It’s a bit hard to argue what is “mainstream” but I would again argue that deep GNNs are not commonplace because the most commonly used datasets don’t necessitate them.

* The datasets used in the evaluation are very limited. Generalizing from these datasets to GNNs as a whole is a bit of a stretch.

* While the paper demonstrates that deep MLPs are problematic and part of the problem, it doesn't convincingly show that aggregation also causes problems. Figure 2 only shows the results for randomly initialized GNNs, which is only mentioned deep in the Appendix as far as I can tell. This fact should be more explicit and it should be honestly discussed what this means for the results and takeaways drawn from Figure 2.


The paper does not show a new position that would be impactful for the research community as several papers have already made the same argument. The new empirical evidence is weaker in comparison and doesn’t provide meaningful new insights. Especially because the most relevant related work is not discussed, my recommendation is a reject.




[0] Arroyo, Á., Gravina, A., Gutteridge, B., Barbero, F., Gallicchio, C., Dong, X., ... & Vandergheynst, P. (2025). On vanishing gradients, over-smoothing, and over-squashing in gnns: Bridging recurrent and graph learning. arXiv preprint arXiv:2502.10818. NeurIPS 2025

[1] Arnaiz-Rodriguez, A., & Errica, F. (2025). Oversmoothing, oversquashing, heterophily, long-range, and more: Demystifying common beliefs in graph machine learning. arXiv preprint arXiv:2505.15547.

**Support:**

3

---

> ### Author Rebuttal · Authors · 2026-03-31
>
> We sincerely thank the Reviewer for their rigorous and critical review. Below, we address each of the Reviewer's concerns in detail.
>
> > w1, w7, q1. Regarding novelty and related works [0, 1]
>
> We first thank the Reviewer for pointing out these relevant recent works [0] and [1]. We will include a detailed discussion of them in our revised manuscript to better contextualize our position.
>
> We consider [0] a valuable work whose findings serve as a complementary counterpart to our position, and believe the juxtaposition of [0] and our work offers a necessary and healthy debate for the field.
> However, we respectfully disagree that [0] has already shown our position and provided the same evidence.
> While both [0] and our work identify convergence of representations toward a zero-vector and link it to oversmoothing, our analysis on its cause and the role of aggregation are fundamentally different.
> * **Opposing Views on Aggregation’s Effect**: [0] argues that aggregation’s norm-contracting nature causes GNNs to suffer from more severe vanishing gradients than MLPs. Conversely, our Figure 1 provides empirical evidence that GNNs are more robust to increased depth than their MLP counterparts.
> * **Misattribution of Zero-Collapsing**: [0] blames the "norm-contracting nature of GNN updates" (i.e., aggregation) for feature collapse. However, our Figure 2 demonstrates that this "zero-collapsing" is primarily driven by transformation and activation steps and occurs equally in MLPs without any aggregation.
>
> Regarding [1], while it offers a valuable demystification of common beliefs, our contribution is distinct in its systematic deconstruction of the evidentiary foundations behind the "aggregation-as-obstacle" narrative. For a detailed comparison and the significance of this distinction, please refer to our response to Reviewer eAFA (w1).
>
> > w2, w4, q2. On Deep GNN Utility and Evidence
>
> We clarify that we do not advocate for universal deep GNN deployment (w2); we explicitly recommend deep GNNs only for specific tasks where they have demonstrated effectiveness, and acknowledge that they may not be beneficial in all settings (lines 334–344).
>
> However, we disagree with the claim that deep GNNs are underused simply because they are "unnecessary" (w4). Even in tasks where depth is beneficial, shallow architectures are consistently adopted. We provide the following examples:
> * Dwivedi et al., (2022): This work proposes datasets require capturing long-range dependencies, yet the maximum GNN depth was set to 5, far below the graph diameter of ~57
> * Platonov et al. (2023): This work proposes heterophilic benchmarks where long-range propagation is critical, yet depth was capped at 5 despite graph diameters reaching 6,824.
> Notably, subsequent works cited in our paper achieved state-of-the-art results on these very datasets by applying significantly deeper GNNs.
>
> In sum, we do not claim that deeper GNNs would always be beneficial (q2). Instead, we argue that even when depth is beneficial, exploration remains limited by the prevailing negative narrative. We call for research to rigorously study when and why depth helps, rather than defaulting to shallow architectures based on an incomplete understanding.
>
> > w3. Missing Purple Lines
>
> The purple line ($\sigma+A$) in the first row of Figure 2 is not missing but overlaps with the red line ($A$), as both exhibit similarly low decay rates. We will add a clarifying note in the revised manuscript to prevent any confusion.
>
> > w5. Dataset Limitations
>
> We have provided a detailed response to Reviewer Pwco (c1). We kindly invite the Reviewer to refer to that response.
>
> > w6. Transparency in Experimental Setup
>
> We emphasize that our analysis of initialized models is not a hidden detail but an explicit and integral part of our research design, as clearly stated in the main text.
> * **The Intentional Research Focus**: The central question of Section 3.3 is explicitly: 'what if oversmoothing occurs so rapidly in initialized networks that optimization cannot even begin?' (lines 217–219). To address this question, we reproduced and further analyzed the experimental results of prior works that utilized initialized models.
> * **Transparent Discussion**: We provide an honest and detailed discussion concluding that the rapid decay is driven by zero-collapsing, a byproduct of Glorot initialization and ReLU activation. We also explicitly frame this as a resolvable initialization issue (lines 275–288).
> * **Further Clarification in Appendix**: Unlike the other work (Oono & Suzuki, 2020; Rusch et al., 2023a) we refer for our experiment, Wu et al. (2023) state their result is obtained from trained network. However, we found that training fails at a depth of 128 layers due to vanishing gradients, leaving the model in its initial state. The mention in the Appendix was specifically intended for further technical clarification of this discrepancy. We will revise the manuscript to ensure this point is stated more clearly.

---

> > ### Author Rebuttal · Reviewer_miFn · 2026-04-02
> >
> > I thank the authors for the rebuttal. I find several concerns unresolved:
> >
> > * The rebuttal's central novelty argument is that [0] "blames the 'norm-contracting nature of GNN updates' (i.e., aggregation) for feature collapse," while this paper shows it is actually zero-collapsing from transformation and activation. However, [0] says the opposite. In its Section 4, [0] explicitly states: "while much of the literature [...] has attributed over-smoothing to iterative aggregation and convergence to a rank-one subspace, our analysis shows that this phenomenon is instead an artifact of representations collapsing to zero." Both papers converge on the same core insight that feature collapse is driven by contractive dynamics, not by aggregation-induced convergence to a rank-one subspace. I acknowledge that the submitted paper provides a more granular component-wise decomposition than [0]. However, this is a difference in granularity, not a "fundamentally different" analysis. [0] further provides a unified theoretical framework, a constructive solution with results on long-range benchmarks, and evidence on trained networks. This is substantially deeper than what this submission provides.
> >
> > * Stripped to its essence, the paper's empirical contribution is: prior oversmoothing studies used Glorot initialization with ReLU, so what they measured as "aggregation-induced oversmoothing" was actually the well-known zero-collapsing phenomenon from the deep learning literature. This is a valid observation, but it is a narrow one, it identifies a confound in prior experimental protocols rather than revealing something new about aggregation's actual role. The theoretical discussion in Section 3.2 re-emphasizes a condition already present in the bound of Oono & Suzuki (2020). The paper highlights an implication of existing theory rather than extending it.
> >
> > * A position paper should ideally open new directions or reframe a field's understanding in a way that generates productive debate. I see limited potential here for two reasons. First, the position ("stop blaming aggregation") is already the conclusion of [0] and partially of [1], so the debate the paper seeks to spark is already underway. Second, the paper's constructive recommendations: deploy deep GNNs where appropriate, study aggregation more carefully, are too general to be actionable and do not follow uniquely from the paper's evidence.
> >
> > I maintain that this work identifies a valid confound in prior oversmoothing experiments, but the same core insight appears in other works with deeper analysis and constructive results (Reviewer eAFA seems to agree on this and provides more references). I do not see sufficient novelty or discussion potential for acceptance.

---

### Official Review · Reviewer_eAFA · 2026-03-13

**Significance:** 1
**Argument Clarity:** 3
**Rating:** 2
**Confidence:** 5

**Questions:**

Please refer to the weaknesses.

**Alternative Views Section:**

Yes

**Compliance With Llm Reviewing Policy A Conservative:**

Affirmed.

**Discussion Potential:**

2

**Paper Summary:**

The paper claims that the aggregation’s negative effects in GNNs have been exaggerated while its benefits have been overlooked. The authors contribute the performance degradation primarily to the transformation layers.

**Position:**

Yes

**Position In Title:**

Yes

**Related Work:**

2

**Strengths And Weaknesses:**

Strengths:

1. The authors review both emperically and theoretical works of oversmoothing in GNNs.

2. The paper is well-written and easy to follow.


Weaknesses:

1. The main issue is that the central claim does not appear to be new. Several papers have already investigated the idea that the performance degradation of deep GNNs may arise from the transformation layers rather than the propagatio, such as [1][2]. For example, experiments similar to those shown in Figure 1 have already been conducted in these prior works.

2. While deep GNNs can be beneficial for certain graph tasks that require modeling long-range dependencies, shallow GNNs already perform well on most commonly used benchmark datasets, especially under label-sparse settings. The paper lacks a clear discussion of when deep GNNs are actually beneficial and what advantages they provide in practice, despite the authors’ suggestion for using deep GNNs.

3. The paper does not provide novel empirical or theoretical findings.

4. The authors also discuss the over-squashing in the introduction section. However, there is no future discussion in the following section.

[1] Zhang, Wentao, et al. "Model degradation hinders deep graph neural networks." Proceedings of the 28th ACM SIGKDD conference on knowledge discovery and data mining. 2022.

[2] Zhou, Kuangqi, et al. "Understanding and resolving performance degradation in deep graph convolutional networks." Proceedings of the 30th ACM international conference on information & knowledge management. 2021.

**Support:**

2

---

> ### Author Rebuttal · Authors · 2026-03-31
>
> We thank the Reviewer for the detailed feedback and for noting that our work (1) thoroughly reviews GNN oversmoothing, (2) is well-written, and (3) effectively organizes the aggregation-centered narrative. We address the concerns below.
>
> > w1. Regarding novelty and related works [1, 2]
>
> While [1, 2] share the insight that aggregation misidentified as the primary cause of GNN degradation, our research offers a fundamentally distinct contribution:
>
> * We propose a novel insight not found in [1, 2]: aggregation may contribute to robustness of performance as depth increases. This discovery stems from our distinct methodology of conducting a direct ablation against MLPs, whereas [1] and [2] merely varied the number of aggregation and transformation within GNNs, failing to isolate the stabilizing effect of aggregation.
>
> * [1] leaves the ‘problematic oversmoothing’ narrative untouched by arguing it occurs at greater depths, while model degradation, their findings, occurs earlier. In contrast, we directly challenge the validity of the oversmoothing narrative itself.
>
> * [2] attributes performance decay to "variance inflammation" and notably blames aggregation for vanishing gradients. Our results directly contradict this: we show that aggregation provides robustness against the decay that occurs in MLPs.
>
> We want to additionally note that, despite these and other observations (as discussed in Section 5.1) challenge the narrative that "GNNs suffer from depth due to aggregation", it remains a dominant research stream. We argue this persistence is because the evidentiary foundations of the narrative, the original experiments and theoretical analyses, had never been directly refuted.
>
> Consequently, unlike [1, 2, 3] (note: [3] is referred to as [1] by Reviewer miFn) or prior works, our paper focuses on rigorously re-examining the original evidence to expose its incompleteness. By directly dismantling these foundations, we:
> * show that aggregation contributes to robustness;
> * clarify that foundational theories do not suggest over-smoothing is inevitable;
> * demonstrate that rapid exponential oversmoothing is driven by transformation-activation dynamics rather than aggregation.
>
> We thus provide a critical, novel perspective that prior literature, including [1, 2, 3], has not sufficiently addressed.
>
> > w2. Utility and Advantages of Depth
>
> Section 4.1 details that depth is essential for tasks requiring high model capacity, large receptive fields, and global structure capture, such as large-scale datasets, long-range dependencies, and heterophilic benchmarks.
>
> While we acknowledge that precisely when deep GNNs are beneficial remains under-explored, our paper argues that the "aggregation-as-obstacle" narrative has been the primary barrier to such discovery. By dismantling the narrative, we enable researchers to rigorously investigate aggregation's actual effects, fostering a clearer community-wide understanding of when and why depth is beneficial.
>
> > w3. Novel empirical or Theoretical Findings.
>
> We believe the Reviewer’s concern stems from applying main track papers’ criteria to a position paper, which is intended to challenge prevailing perspectives. As stated in the position paper CFP: 'Position papers make an argument for a viewpoint or perspective about what should be done, in contrast to main track papers, which report on advances that have already been accomplished'.
>
> However, even by rigorous standards, our paper provides pivotal, previously unreported empirical findings that directly support our novel position:
>
> * **Empirical Robustness**: We demonstrate that GNNs are more robust to depth than their MLP counterparts (Figure 1), directly contradicting the "aggregation-as-obstacle" narrative.
>
> * **Discovery of Zero-Collapsing**: We prove that the rapid decay often blamed on aggregation is actually driven by zero-collapsing from transformation and activation dynamics (Figure 2), a crucial distinction never before clarified in GNN literature.
>
> These findings are not merely "supporting data" but foundational discoveries that dismantle a dominant misconception. By correcting this misdiagnosis, our work provides an actionable roadmap for the community to shift focus toward deep GNN exploration. We believe that enabling such a fundamental shift in perspective is a key contribution of our position paper.
>
> > w4. Discussion in Oversqushing
>
> Oversquashing was mentioned in the Introduction to provide the broader context of the "aggregation-as-obstacle" narrative. We agree that its absence in subsequent sections may cause confusion. In the revision, we will explicitly delimit our scope to oversmoothing, where we find the evidentiary foundations most in need of re-examination. To better situate our work, we will also add references to recent studies that similarly challenge oversquashing misconceptions, ensuring a clearer distinction between these two phenomena for the readers.

---

> > ### Author Rebuttal · Reviewer_eAFA · 2026-04-01
> >
> > Thanks for the authors rebuttal.
> >
> > I agree that position papers are evaluated differently. However, I believe the position papaers should propose new insights instead of stating the well-known facts again. As state in the ICML Position Paper CFP:
> >
> > > To constitute a proper scientific contribution, a position must be non-obvious, falsifiable and defendable against credible alternatives.
> >
> > There are several works alread seperate the propagation with transfromation in GNNs and study the effect of these two components. The authors still claim "prior work has implicitly blamed aggregation for these challenges without conducting the necessary ablations to isolate message passing from standard MLP dynamics."
> >
> > Therefore, I still doubt the real contribution of this position paper.

---

### Official Review · Reviewer_PA9G · 2026-03-13

**Significance:** 3
**Argument Clarity:** 4
**Rating:** 5
**Confidence:** 2

**Questions:**

* All experiments are performed on text‑attributed graphs. Do the authors expect their claims to generalize to graphs without node attributes (pure structural graphs)? Please clarify or provide experiments/analysis for non‑attributed graphs.

* Evaluation focuses on node classification. Do the authors expect the same conclusions to hold for other graph tasks (e.g., graph classification, link prediction)? If so, please discuss or provide supporting evidence.

**Alternative Views Section:**

Yes

**Compliance With Llm Reviewing Policy A Conservative:**

Affirmed.

**Discussion Potential:**

3

**Final Justification:**

Despite its relatively limited experiments, this position paper provides an important counter-perspective by arguing that the widely-accepted blame on the aggregation mechanism for GNN depth limitations may be overstated. It prompts the community to rethink the fundamental causes of performance degradation in deep GNNs, which I consider a valuable and thought-provoking contribution worthy of acceptance.

**Paper Summary:**

This position paper argues that current research has overly attributed GNNs’ capabilities to their aggregation mechanisms. Through re‑experimentation and theoretical analysis, the authors show that the negative effects of aggregation have been exaggerated while its benefits have been overlooked.

**Position:**

Yes

**Position In Title:**

Yes

**Related Work:**

4

**Strengths And Weaknesses:**

**Strengths**:
* The paper addresses a timely and important topic in GNN research.
* The literature review, theoretical analysis, and empirical ablation studies are comprehensive and well organized, and together provide substantial support for the central claims.


**Weakness: **
The empirical support for the claims relies primarily on node‑distance / similarity measurements used to explain over‑smoothing and to argue that aggregation contributes to robustness. However, high node similarity preserved in deep layers does not necessarily imply improved downstream performance (e.g., node classification). The manuscript should better connect the similarity metrics to task performance, e.g., by showing direct correlations between the proposed metrics and downstream accuracy, or by providing controlled ablations demonstrating when deeper aggregation helps versus hurts task metrics.

**Support:**

3

---

> ### Author Rebuttal · Authors · 2026-03-31
>
> We sincerely thank the Reviewer for their valuable feedback and for recognizing that our work (1) addresses a timely and important topic in GNN research, (2) provides a comprehensive and well-organized literature review and theoretical analysis, and (3) offers substantial empirical support through systematic ablation studies. Below, we address each of the Reviewer's concerns in detail.
>
> > w1. The empirical support for the claims relies primarily on node‑distance / similarity measurements used to explain over‑smoothing and to argue that aggregation contributes to robustness. However, high node similarity preserved in deep layers does not necessarily imply improved downstream performance (e.g., node classification). The manuscript should better connect the similarity metrics to task performance, e.g., by showing direct correlations between the proposed metrics and downstream accuracy, or by providing controlled ablations demonstrating when deeper aggregation helps versus hurts task metrics.
>
> We agree that high node similarity does not necessarily imply improved downstream performance, and we have never claimed otherwise. To clarify, our claim that aggregation contributes to robustness is supported by the accuracy results in Figure 1, not by node similarity measurements.
> The similarity measurements are adopted from prior oversmoothing analyses to reproduce and re-examine their results. Since a similarity measure of zero implies complete convergence of representations, which clearly harms downstream performance, prior work has analyzed how quickly and under what conditions this occurs. Our contribution is to show that this analysis is incomplete.
>
> We also point out in Section 4.2, in line with your insight, that these measures fail to capture harmful smoothing that directly affects performance, and identify this as an important direction for future research.
>
> > q1. All experiments are performed on text‑attributed graphs. Do the authors expect their claims to generalize to graphs without node attributes (pure structural graphs)? Please clarify or provide experiments/analysis for non‑attributed graphs. / q2. Evaluation focuses on node classification. Do the authors expect the same conclusions to hold for other graph tasks (e.g., graph classification, link prediction)? If so, please discuss or provide supporting evidence.
>
> Our choice of node classification on text-attributed graphs follows a consistent, targeted strategy aligned with the nature of a position paper:
> * **Deconstructing the Source**: Our primary goal is to re-examine the specific experiments that established the "aggregation-as-obstacle" narrative. Since that narrative was built almost exclusively on text‑attributed graphs, we focused on these same datasets to prove that the original evidence itself was incomplete. We believe this targeted refutation is a necessary first step before generalizing further.
> * **Expanded Evidence**: However, we recognize that the papers we reference in relation to Figures 2 and 3 report results on a wider range of datasets (randomly generated Erdős–Rényi graphs in Oono & Suzuki, 2020; and Cora, CiteSeer, PubMed, Cornell, Texas, and Wisconsin in Wu et al.,2023; Texas, Cora, and Facebook (Cornell5) in Rusch et al., 2023a). To address this, we provide additional results on the real-world datasets used in these works in the [anonymous link](https://anonymous.4open.science/api/repo/ICML2026_Rebuttal-5A23/file/_ICML_2026__Oversmoothing_Rebuttal.pdf?v=2c5cc4c4). Note that Facebook (Cornell5) was excluded due to GPU memory constraints arising from its large scale (4.8M nodes). **Across all additional benchmarks, the results consistently confirm our finding**: the rapid decay is dominated by the combination of transformation and activation, not aggregation.
> * **Generalizability and Future Directions**: Our discovery in Figure 2 that decay is driven by zero-collapsing is a fundamental property of any architecture incorporating MLP structures. Therefore, zero-collapsing is not inherently limited to node attributes or a specific task. In contrast, in the case of the findings in Figure 1, the empirical manifestation may vary across non-attributed graphs or different tasks (e.g., link prediction). However, providing a general theory is not the goal of our paper. Consistent with the role of a Position Paper, our goal is to clear the ground for new theories rather than presenting a finished one ourselves. To this end, instead of a single alternative framework, we propose a set of critical research directions in Section 4.2 that can now be rigorously pursued on this corrected evidentiary foundation. We believe identifying what needs to be studied next is the most constructive contribution a position paper can offer to the community.

---

> > ### Author Rebuttal · Reviewer_PA9G · 2026-04-06
> >
> > Thank you to the authors for the clarification, which has  resolved my concern. I'm happy to raise my score accordingly.

---

### Official Review · Reviewer_Pwco · 2026-03-13

**Significance:** 3
**Argument Clarity:** 3
**Rating:** 4
**Confidence:** 4

**Questions:**

See the Weakness.

**Alternative Views Section:**

Yes

**Compliance With Llm Reviewing Policy A Conservative:**

Affirmed.

**Discussion Potential:**

3

**Paper Summary:**

This position paper revisits the dominant claim that aggregation is the main cause of depth-related degradation in GNNs. It argues that prior evidence is incomplete, presents ablations comparing GCN/GAT with MLPs across depth, reinterprets oversmoothing theory as conditional rather than inevitable, and claims that rapid exponential collapse is driven largely by transformation and activation rather than aggregation itself.

**Position:**

Yes

**Position In Title:**

Yes

**Related Work:**

3

**Strengths And Weaknesses:**

Pros:
- The paper tackles a fundamental and highly relevant question in graph representation learning, namely whether aggregation has been overly blamed for the failure of deep GNNs.
- A key strength of the work is its effort to challenge a widely accepted narrative through relatively clean and interpretable comparisons, especially the depth-wise analysis between GNNs and corresponding MLP variants.
- The paper is well organized and generally well written. The main argument is presented clearly, and the discussion succeeds in connecting empirical findings, theoretical intuition, and broader implications for deep GNN design.

Cons:
- The paper advances a strong and timely position, but the empirical evidence directly supporting its central claim is still somewhat limited in scope. Most of the key analyses are conducted on standard citation benchmarks and a relatively small set of canonical architectures, which leaves some uncertainty about how broadly the conclusions transfer to more challenging graph settings.
- The theoretical discussion is insightful and helps clarify why prior oversmoothing arguments may be incomplete. However, it is more clarificatory than constructive: the paper effectively questions an existing explanation, but does not yet provide an equally strong alternative theoretical framework.

**Support:**

3

---

> ### Author Rebuttal · Authors · 2026-03-31
>
> We sincerely thank the Reviewer for the thoughtful evaluation and are deeply encouraged by the assessment that our paper advances a strong and timely position. We especially appreciate the recognition that our work (1) challenges the prevailing narrative on aggregation, (2) provides clean GNN-MLP comparisons, and (3) clarifies theoretical frameworks. Below, we address each concern in detail.
>
> > c1. The paper advances a strong and timely position, but the empirical evidence directly supporting its central claim is still somewhat limited in scope. Most of the key analyses are conducted on standard citation benchmarks and a relatively small set of canonical architectures, which leaves some uncertainty about how broadly the conclusions transfer to more challenging graph settings.
>
> We first thank the Reviewer for the encouraging remarks on the timeliness and strength of our position. To clarify the empirical scope, we highlight that our choice of datasets follows a consistent, targeted strategy designed for a position paper.
>
> * **Deconstructing the Source**: Our primary goal is to re-examine the specific experiments that established the "aggregation-as-obstacle" narrative. Since that narrative was built almost exclusively on canonical citation benchmarks, we focused on these same datasets to prove that the original evidence itself was incomplete. We believe this targeted refutation is a necessary first step before generalizing further.
>
> * **Expanded Evidence**: However, we recognize that the papers we reference in relation to Figures 2 and 3 report results on a wider range of datasets (randomly generated Erdős–Rényi graphs in Oono & Suzuki, 2020; and Cora, CiteSeer, PubMed, Cornell, Texas, and Wisconsin in Wu et al.,2023; Texas, Cora, and Facebook (Cornell5) in Rusch et al., 2023a). To address this, we provide additional results on the real-world datasets used in these works in the [anonymous link](https://anonymous.4open.science/api/repo/ICML2026_Rebuttal-5A23/file/_ICML_2026__Oversmoothing_Rebuttal.pdf?v=2c5cc4c4). Note that Facebook (Cornell5) was excluded due to GPU memory constraints arising from its large scale (4.8M nodes). **Across all additional benchmarks, the results consistently confirm our finding**: the rapid decay is dominated by the combination of transformation and activation, not aggregation.
>
> > c2. The theoretical discussion is insightful and helps clarify why prior oversmoothing arguments may be incomplete. However, it is more clarificatory than constructive: the paper effectively questions an existing explanation, but does not yet provide an equally strong alternative theoretical framework.
>
> We agree that our theoretical discussion is primarily clarificatory. However, we argue that clarification is a constructive act when it dismantles the dominant misconceptions that have restricted the field's exploration.
>
> Consistent with the role of a Position Paper, our goal is to clear the ground for new theories rather than presenting a finished one ourselves. To this end, instead of a single alternative framework, we propose a set of critical research directions in Section 4.2 that can now be rigorously pursued on this corrected evidentiary foundation. We believe identifying what needs to be studied next is the most constructive contribution a position paper can offer to the community.

---

### Decision · Program_Chairs · 2026-04-30

**Decision:**

Reject

**Comment:**

Some reviewers were very favorable toward this work. Oversmoothing and oversquashing play an important role in discussions about graph learning, and as such, new light on these topics could create great benefits. At the same time, other reviewers argued that the position presented in this work is not new. Therefore, if the position is not new, it is expected that the position paper will contain some novelty in terms of new argumentation or new evidence to support this position, given that the position itself is not new.

The issue of novelty was discussed in this review session. Reviewer eAFA questioned the novelty of this work, noting that previous studies have already made similar claims and pointing to relevant references. Reviewer miFn questioned the quality of the evaluation presented in this work. The authors debated these arguments, focusing on the different objectives and criteria for position papers versus research papers. Indeed, the novelty requirements for a position paper are different from the requirements for a research paper. While the references provided imply that aggregation is not to be blamed, it is still a common belief in the field, and therefore there is a benefit to a paper that states this argument in its title. At the same time, it could be argued that since the opinion presented here is already present in prior art, why it would create an impact that previous papers could not.

The position paper, as is, can draw some attention, but this may be marginal. I strongly encourage the authors to continue this work and add additional support for their position.